# Real-Time PCR Assay for the Detection and Quantification of Roe Deer to Detect Food Adulteration—Interlaboratory Validation Involving Laboratories in Austria, Germany, and Switzerland

**DOI:** 10.3390/foods10112645

**Published:** 2021-11-01

**Authors:** Barbara Druml, Steffen Uhlig, Kirsten Simon, Kirstin Frost, Karina Hettwer, Margit Cichna-Markl, Rupert Hochegger

**Affiliations:** 1Department of Molecular Biology and Microbiology, Institute for Food Safety Vienna, Austrian Agency for Health and Food Safety (AGES), Spargelfeldstraße 191, 1220 Vienna, Austria; barbara.druml@univie.ac.at; 2Department of Analytical Chemistry, Faculty of Chemistry, University of Vienna, Währinger Straße 38, 1090 Vienna, Austria; 3QuoData GmbH, Prellerstraße 14, 01309 Dresden, Germany; steffen.uhlig@quodata.de (S.U.); kirsten.simon@quodata.de (K.S.); kirstin.frost@quodata.de (K.F.); karina.hettwer@quodata.de (K.H.)

**Keywords:** real-time PCR, roe deer, game meat, detection, quantification, food authentication, validation, interlaboratory ring trial, probability of detection

## Abstract

Game meat products are particularly prone to be adulterated by replacing game meat with cheaper meat species. Recently, we have presented a real-time polymerase chain reaction (PCR) assay for the identification and quantification of roe deer in food. Quantification of the roe deer content in % (*w*/*w*) was achieved relatively by subjecting the DNA isolates to a reference real-time PCR assay in addition to the real-time PCR assay for roe deer. Aiming at harmonizing analytical methods for food authentication across EU Member States, the real-time PCR assay for roe deer has been tested in an interlaboratory ring trial including 14 laboratories from Austria, Germany, and Switzerland. Participating laboratories obtained aliquots of DNA isolates from a meat mixture containing 24.8% (*w*/*w*) roe deer in pork, roe deer meat, and 12 meat samples whose roe deer content was not disclosed. Performance characteristics included amplification efficiency, level of detection (LOD_95%_), repeatability, reproducibility, and accuracy of quantitative results. With a relative reproducibility standard deviation ranging from 13.35 to 25.08% (after outlier removal) and recoveries ranging from 84.4 to 114.3%, the real-time PCR assay was found to be applicable for the detection and quantification of roe deer in raw meat samples to detect food adulteration.

## 1. Introduction

Game meat is appreciated because of its characteristic sensory properties, especially its distinct flavor and tenderness. In general, game meat is regarded as healthier than meat from domestic species due to its lower intramuscular fat and cholesterol content and its high content of polyunsaturated fatty acids [1]. Like other commercial food products, game meat products must comply with national and international food legal regulations. Hence, game meat products have to be not only safe but also authentic. Adulteration of meat products by complete or partial replacing of more expensive meat with cheaper meat species is, however, known to be a global issue [2,3,4,5]. Due to its high price and seasonal availability, game meat is particularly prone to this kind of adulteration.

Fraudulent labeling of game meat products can only be detected by applying specific and sensitive analytical methods. Both conventional polymerase chain reaction (PCR) and real-time PCR assays have been developed for the detection of a variety of game meat species in food [6,7,8,9,10]. According to the Codex Alimentarius Austriacus, a collection of standards and product descriptions serving as guidelines for food inspectors, the declaration “game sausage” may only be used if ≥38% of the total meat content in the sausage originates from game species [11]. Thus, methods providing quantitative information are required in addition to qualitative methods for game meat authentication. 

Recently, we have developed real-time PCR assays for the detection and quantification of red deer, sika deer, fallow deer, roe deer, and wild boar in food [12,13,14,15,16,17,18,19,20]. Quantification of meat species in food by real-time PCR is a challenging task [2,21]. The main difficulty arises from the necessity to correlate the DNA concentration determined by real-time PCR to the meat content given in weight/weight (*w*/*w*). Factors such as tissue type, the number of cells per unit of mass, genome size, and DNA extractability may affect the accuracy of quantitative results [21]. Since the number of mitochondrial DNA copies varies between different animal species and tissue types, we have designed primers targeting single copy genes [12,14,18]. In order to compensate for differences in tissue composition, we pursued a relative quantification approach. In addition to the game species-specific real-time PCR assay, DNA isolates were subjected to a reference real-time PCR assay. The reference real-time PCR assay allows amplification of a conserved 97 bp fragment of the myostatin gene in mammalian and poultry species [22]. Relative quantification is less labor intensive than quantification by using matrix-specific calibrators, another quantification strategy applied in meat species authentication [23,24].

Roe deer (*Capreolus capreolus*) is one of the most frequently consumed deer species in Europe. The real-time PCR assay for roe deer developed recently targets a 62 bp sequence of the roe deer lactoferrin gene [14]. The real-time PCR assay did not show cross-reactivity with 23 animal and 43 plant species tested. An increase in the fluorescence signal was only observed for fallow deer. Since the difference of Ct values between roe deer and fallow deer was >13, low cross-reactivity was considered negligible. In order to investigate whether the real-time PCR is fit for its intended purpose [25], it was subjected to in-house validation, including determination of amplification efficiency, level of detection (LOD), limit of quantification (LOQ), repeatability, and robustness. In-house validation data suggested that the real-time PCR assay for roe deer is suitable for routine analysis. However, for method standardization, evaluation of interlaboratory variability is a prerequisite [25].

Aiming at harmonizing analytical methods for food authentication across EU Member States, in 2017 the real-time PCR assay for roe deer was tested in an interlaboratory ring trial on behalf of the Federal Office for Consumer Protection and Food Safety, Berlin, Germany for the Official Collection of Methods ASU § 64 LFGB. Performance characteristics included amplification efficiency, LOD_95%_, repeatability, reproducibility, and accuracy of quantitative results. Results of interlaboratory validation of the real-time PCR assay for roe deer are summarized in this paper. 

## 2. Materials and Methods

### 2.1. Participating Laboratories

The interlaboratory ring trial was organized by the Austrian Agency for Health and Food Safety (AGES) on behalf of the Federal Office for Consumer Protection and Food Safety in Germany. The following laboratories participated in the ring trial (in alphabetical order): Austrian Agency for Health and Food Safety (AGES), Vienna, Austria; Cantonal Office of Consumer Protection Aargau, Aarau, Switzerland; Chemical and Veterinary Investigation Office Freiburg, Freiburg, Germany; Chemical and Veterinary Analytical Institute Muensterland-Emscher-Lippe, Muenster, Germany; German Federal Institute for Risk Assessment (BfR), Berlin, Germany; Impetus GmbH & Co. Bioscience KG, Bremerhaven, Germany; Institute of Hygiene and Environment, Hamburg, Germany; Max Rubner-Institut, Kulmbach, Germany; Official Food Control Authority of the Canton Zurich, Zurich, Switzerland; Saxon State Institute of Health and Veterinary Affairs, Dresden, Germany; Saxony-Anhalt State Office for Consumer Protection, Halle, Germany; State Laboratory Berlin-Brandenburg, Berlin, Germany; State Office Laboratory Hessen, Gießen, Germany; State Office of Agriculture, Food Safety and Fisheries Mecklenburg-Vorpommern, Rostock, Germany. 

### 2.2. Design of the Interlaboratory Ring Trial

The design of the interlaboratory ring trial is outlined in Figure 1. Participants obtained an aliquot of a DNA isolate from a meat mixture containing 24.8% (*w*/*w*) roe deer in pork, an aliquot of a DNA isolate from roe deer meat, and coded aliquots of DNA isolates from 12 meat samples.

The DNA isolate from the meat mixture containing 24.8% (*w*/*w*) roe deer in pork was used for calibration of the roe deer real-time PCR assay and the reference real-time PCR assay. From the slope of the calibration curves, the amplification efficiency was calculated. The isolate had a DNA concentration of 20 ng/µL and contained 1440 copies of the roe deer specific target sequence per µL. The DNA isolate was serially diluted with bidistilled water (ddH_2_O) to obtain DNA isolates with a concentration of 5, 1.25, 0.3125, and 0.078 ng/µL, corresponding to 360, 90, 22.5, and 5.625 copies of the roe deer-specific target sequence per µL, respectively. The diluted DNA isolates were analyzed by the roe deer real-time PCR assay and the reference real-time PCR assay in two PCR replicates each.

The DNA isolate from roe deer meat served for determination of LOD_95%_ of the roe deer real-time PCR assay. The DNA isolate containing 5000 copies of the roe deer-specific target sequence per 5 µL was serially diluted with a buffer containing herring sperm DNA (20 ng/µL; also provided by the organizer of the ring trial). DNA isolates containing 500, 20, 10, 5, 2, 1, 0.5, or 0.1 copies of the roe deer-specific target sequence per 5 µL were prepared. DNA isolates containing ≤ 20 copies of the roe deer specific target sequence per 5 µL were analyzed by the roe deer real-time PCR assay in six PCR replicates. Herring sperm DNA was used as no template control (NTC, two PCR replicates).

DNA isolates from 12 meat samples (Table 1) served for determination of the applicability of the roe deer real-time PCR assay for providing quantitative results. Participants directly analyzed the DNA isolates by the roe deer real-time PCR assay and the reference real-time PCR assay in three PCR replicates each.

### 2.3. Meat Samples 

Meat samples included nine meat mixtures and three sausages (Table 1). Meat mixtures were prepared at the AGES. Fresh roe deer and pork meat was taken in a slaughterhouse by a food inspector. After cutting and homogenizing roe deer and pork meat in a cutter (robot coupe R5 plus, Toperczer, Schwechat-Rannersdorf, Austria) for 5–10 min, at least 2 kg of mixtures were prepared by weighing out the respective amounts of meat and homogenizing the mixture in the cutter. With the exception of meat mixture 1 (meat sample 1), which was free of roe deer, meat mixtures contained roe deer in the range from 1 to 49.4% (*w*/*w*). Meat mixture 9 (meat sample 9) was boiled at 100 °C for 20 min. Immediately after preparation, meat mixtures were subjected to DNA isolation.

Sausage 1 (meat sample 10) was a model sausage, containing 21.0% roe deer. The other two sausages (meat sample 11 and 12) were purchased from a supermarket, with meat sample 11 being brewed and meat sample 12 being a raw sausage. Both sausages were declared to contain roe deer in the range from 5 to 10% (*w*/*w*). After homogenization, sausages were stored at −20 °C until DNA isolation.

Participants directly subjected DNA isolates to real-time PCR analysis.

### 2.4. Isolation of Genomic DNA

Isolation of genomic DNA from meat mixtures, sausages, and roe deer meat was carried out at the AGES by applying the official method L 00.00–119 [26]. After isolating genomic DNA twice, the undiluted DNA isolates were combined. 

DNA concentration of the (combined) DNA isolate from the meat mixture containing 24.8% (*w*/*w*) roe deer in pork (used for calibration of the roe deer real-time PCR assay and the reference real-time PCR assay) was adjusted to 20 ng/μL. The copy number of the roe deer specific target sequence per 5 µL, determined by droplet digital PCR (ddPCR, QX200 Droplet Generator, QX200 Droplet Reader (Bio-Rad, Hercules, CA, USA)), was 1440 copies/µL.

After determining the copy number of the roe deer-specific target sequence in the DNA isolate from roe deer meat (serving for determination of LOD_95%_) by ddPCR, the DNA isolate was diluted to obtain 1000 copies/µL.

### 2.5. Real-Time PCR

Sequences and concentrations of primers and probes for the roe deer real-time PCR assay and the reference real-time PCR assay are given in Table 2. Primers and probes were provided by the AGES. All participants used the QuantiTect Multiplex PCR-Kit (NoROX, Qiagen, Hilden, Germany).

Real-time PCR reactions were performed in a total volume of 25 µL, consisting of 20 µL of reaction mix and 5 µL of DNA isolate. The following temperature program was applied for both the roe deer real-time PCR assay and the reference real-time PCR assay: 15 min at 95 °C, 45 cycles of 60 s at 94 °C and 60 s at 60 °C. 

### 2.6. Data Evaluation and Statistical Analysis

For each laboratory, the amplification efficiency, E, was calculated from the slope of the standard curve: $E\left(\%\right)=\left(10^{\frac{-1}{\mathrm{slope}}}-1\right)\;\cdot \;100$.

The probability of detection across laboratories, POD, was calculated as follows:

POD (x) = 1 − exp (−λ_o_ · x^b^) with λ_o_ being the average amplification probability and b being the slope across laboratories. Here, both parameters λ_o_ and b were estimated based on a generalized linear mixed model as described in Uhlig et al. [27].

LOD_95%_ based on the POD curve was calculated as
LOD_95%_ = (−ln(0.05)/λ_o_)^1/b^

The content of roe deer meat in relation to the total meat content of the meat sample was calculated as follows:
$$\mathrm{concentration}\mathrm{of}\mathrm{roe}\mathrm{deer}\mathrm{DNA}\;(\mathrm{ng}/µL)\;=\;10^{\frac{{\mathrm{Ct}}_{\mathrm{spec}}-d_{\mathrm{spec}}}{{\mathrm{slope}}_{\mathrm{spec}}}}$$
$$\mathrm{concentration}\mathrm{of}\mathrm{total}\mathrm{meat}\mathrm{DNA}\;(\mathrm{ng}/µL)\;=\;10^{\frac{{\mathrm{Ct}}_{\mathrm{ref}}-d_{\mathrm{ref}}}{{\mathrm{slope}}_{\mathrm{ref}}}}\;$$
with _spec_ and _ref_ referring to the roe deer real-time PCR assay and the reference real-time PCR assay, respectively
Ct: Ct valued: intercept of the standard curveslope: slope of the standard curve
$$\mathrm{roe}\mathrm{deer}\mathrm{meat}\mathrm{content}\;(\%)\;=\;\frac{\mathrm{concentration}\;\mathrm{of}\;\mathrm{roe}\;\mathrm{deer}\;\mathrm{DNA}\;\left(\mathrm{ng}/µL\right)}{\mathrm{concentration}\;\mathrm{of}\;\mathrm{total}\;\mathrm{meat}\;\mathrm{DNA}\;\left(\mathrm{ng}/µL\right)}\cdot 100$$

Repeatability, reproducibility, and accuracy of the roe deer meat content were determined using the statistical approaches according to ISO 5725-2 [28] as well as according to the specifications of the ASU § 64 LFGB [29].

Statistical analyses were performed by QuoData GmbH using the software package PROLab Plus [30]. Results were subjected to several outlier tests to check for outliers. The presence of outliers within the laboratories was tested as well as whether the variances of the laboratories were approximately the same and whether systematic errors affected the mean values.

## 3. Results and Discussion

### 3.1. Amplification Efficiency

Table 3 summarizes the slope of the laboratory-specific standard curve, coefficient of determination (R^2^), and amplification efficiency (E) for both the roe deer real-time PCR assay and the reference real-time PCR assay, obtained by analyzing serially diluted DNA isolates from a meat mixture containing 24.8% (*w*/*w*) roe deer in pork in two PCR replicates each.

In case of laboratory 11, the slope of the standard curve (Grubbs test, α = 0.05) and the amplification efficiency (Grubbs test, α = 0.01) obtained for the roe deer real-time PCR assay as well as the coefficient of determination obtained for the reference real-time PCR assay (Grubbs test, α = 0.01) were identified as outliers.

According to the guidelines recommended by the European Network of GMO (Genetically Modified Organisms) Laboratories (ENGL) [31], the slope should be between −3.1 and −3.6, corresponding to an amplification efficiency of ~90 to 110%. In almost all cases, slope and amplification efficiency were within the recommended range, with the exception of some values from laboratory 4 and 11. The coefficient of determination, R^2^, is recommended to be >0.98 [31]. All laboratories fulfilled this criterion, with the exception of laboratory 11.

### 3.2. Level of Detection (LOD_95%_)

Table 4 summarizes the laboratory-specific number of positive results obtained by repeated analysis (six PCR replicates) of a DNA isolate from roe deer meat, diluted to 20 to 0.1 copies of the roe deer specific target sequence per 5 µL. Table 5 gives the number of positive results obtained for each dilution step in relation to the total number of tests (*n* = 84). Down to a copy number of five copies per 5 µL, all tests resulted in an increase in the fluorescence signal within 45 cycles. For 2, 1, 0.5, and 0.1 copies per 5 µL, the percentage of positive results was decreased to 86.9, 75.0, 50.0, and 10.7%, respectively.

#### 3.2.1. LOD_95%_ According to Simplified Calculation Approaches

In GMO analysis, LOD_95%_ of a real-time PCR assay is defined as the lowest copy number of a target DNA sequence in a sample, for which a positive result is obtained with a detection probability, *p*, of 95% (LOD_95%_). We used three simplified calculation approaches for the determination of LOD_95%_. In the first approach, LOD_95%_ was regarded as the lowest copy number for which all replicates in all laboratories were positive. In the second approach, LOD_95%_ was considered the lowest copy number of the roe deer-specific target sequence, for which the lower limit of the 90% confidence interval for the detection probability *p*, *p*_u_, was achieved with a probability ≥95%. In the third approach, LOD_95%_ was defined as the lowest copy number of the target sequence, for which ≥95% of the tests yielded a positive result. With all three calculation approaches, LOD_95%_ of the real-time PCR assay for roe deer was determined to be five copies of the roe deer specific target sequence per 5 µL (Table 5).

#### 3.2.2. LOD_95%_ Derived from the Mixed Model for the POD Curve

In addition, we determined LOD_95%_ by applying a statistical model for calculating the probability of detection (POD) across laboratories. Since its introduction by Uhlig et al. [27], this model has already been used several times to determine the sensitivity of real-time PCR assays [32,33,34,35,36,37]. Qualitative results obtained for the seven dilution steps of the DNA isolate from roe deer meat were used to determine the laboratory standard deviation σ_L,_ and the LOD_95%_ for the median laboratory, as described previously [27]. Table 6 summarizes the model parameters, including the estimated values for the average amplification probability λ_o_, and the slope b for describing the POD curve across laboratories in dependence of the copy number of the target sequence (Figure 2). σ_L_ was determined to be 0.15, and the LOD_95%_ for the median laboratory 2.4 copies of the target sequence per 5 µL.

Figure 2 shows the POD curve across laboratories together with the 95% confidence and prediction range as well as the laboratory-specific rates of detection (ROD) with the respective 90% confidence range. In addition, the ideal POD curve obtained under optimal conditions is given.

The POD curve across laboratories (dark blue) was found to lie above the ideal curve obtained under optimal conditions (dashed), which would mean that the obtained LOD_95%_ is better than theoretically achievable. The difference between both curves was statistically significant (*p* < 0.05), suggesting that the actual copy numbers were at least 1.05-fold higher than the nominal copy numbers in the diluted DNA isolates. However, by taking a standard measuring uncertainty of 10% of the DNA isolate from roe deer meat into account, the difference can be considered statistically insignificant.

### 3.3. Analysis of Meat Samples

DNA isolates from meat samples (meat samples 1–12, Table 1) were analyzed by the roe deer real-time PCR assay and the reference real-time PCR assay in three PCR replicates each.

#### 3.3.1. False Positive and False Negative Results

Results obtained with the roe deer real-time PCR assay for the meat mixture that did not contain roe deer (meat sample 1) and samples containing roe deer (meat samples 2–12) were used to determine the rate of false positive and false negative results, respectively. Analysis of 12 meat samples in PCR triplicates in 14 laboratories resulted in a total of 504 results; 42 thereof were obtained for meat sample 1 and 462 for meat samples 2–12, containing roe deer. The qualitative result was correct for all meat samples, there were neither false positive nor false negative results.

#### 3.3.2. Quantitative Results

Evaluation of quantitative results was based on results obtained for meat samples 2–12. Meat sample 1 was not taken into account since it did not contain roe deer. The roe deer content in % (*w*/*w*) was calculated by relating the DNA concentration (ng/µL) determined by the roe deer real-time PCR assay to the DNA concentration (ng/µL) determined by the reference real-time PCR assay.

Single outliers within one laboratory, detected for four samples in four different laboratories, were removed first. Furthermore, results for three meat samples (6, 8, and 12) obtained by one laboratory each show a statistically significantly excessive variance of the triplicates. Statistical evaluation according to ASU § 64 LFGB (based on ISO 5725-2) [28,29] was based on the data after outlier elimination. Table 7 gives the statistical parameters for the determination of the roe deer content (%) in the 11 meat samples containing roe deer. Reproducibility standard deviation, s_R_, is a measure for the variability between laboratories, whereas the repeatability standard deviation, sr, characterizes the variability within a laboratory under repeatable conditions. Based on reproducibility and repeatability standard deviation, the reproducibility limit, R, and repeatability limit, r, were calculated. Reproducibility and repeatability limits are a measure of the maximally expected deviation between two values obtained for a specific sample in different laboratories and in the same laboratory, respectively.

Relative repeatability standard deviation ranged from 6.60% (sample 8) to 17.71% (sample 2), and relative reproducibility standard deviation from 13.35% (sample 8) to 30.22% (sample 5). The rather high relative reproducibility standard deviation obtained for sample 5 decreased to 21.42% when results obtained by laboratory 11 were not taken into account. According to the ENGL guidelines, relative repeatability and relative reproducibility standard deviations should be <25 and <35%, respectively [31]. The roe deer real-time PCR assay fulfilled these criteria and can therefore be considered suitable for achieving reproducible results.

The aim of the ring trial was to validate the real-time PCR assay for roe deer. The suitability of the DNA isolation method (official method L 00.00–119) has been demonstrated before. Thus, the participants did not have to isolate DNA from the samples. DNA isolates, prepared at the AGES, were provided by the organizer of the ring trial. The relative reproducibility standard deviation given above therefore does not include variability caused by DNA isolation. Furthermore, all participants of the ring trial used the same PCR kit (QuantiTect Multiplex PCR-Kit). In principle, the use of different PCR kits might result in higher relative reproducibility standard deviation than the value given above. However, in a preliminary experiment, PCR kits from different providers did not lead to significantly different results.

In addition to data on the repeatability and reproducibility, Table 7 contains recoveries obtained for meat samples 2 to 10. For meat samples 2 to 8 and 10, recovery ranged from 84.4 to 114.3%. With 528.2%, recovery obtained for sample 9 was drastically too high. Sample 9 was the only meat mixture that had been heat-treated (boiled at 100 °C for 20 min). Quantification of the meat content in heat-treated foods by real-time PCR is known to be challenging [38,39,40,41,42]. In several studies, DNA isolates from heat-treated food products were found to yield higher Ct values than DNA isolates from untreated ones [40,41,42]. The five-fold overestimation of the roe deer content for sample 9 can be explained by differences in the amplifiability of the reference sequence compared to the roe deer-specific sequence. With the referenced real-time PCR assay, higher Δ Ct values (difference in the Ct values obtained for DNA isolates from raw and heat-treated samples) were obtained than with the roe deer real-time PCR assay. This result suggests that the reference real-time PCR assay is not applicable for heat-treated meat mixtures. With an amplicon length of 97 bp, the amplicon was substantially longer than that obtained with the roe deer real-time PCR assay (62 bp). We assume that an alternative reference real-time PCR assay published recently [43] is more suitable for heat-treated meat mixtures since it results in a 70 bp amplicon.

For interlaboratory evaluation, combination scores of systematic deviations, RSZ (rescaled sum of z_U_ scores), and relative laboratory performance, RLP, [44] across all samples were used. RSZ is based on a standardized sum of all z_U_ scores (corrected z scores), measuring the deviations of the mean value of a laboratory from the total mean value. If the RSZ is within −2 and +2, the respective laboratory does not show a significant systematic deviation. RLP is ideally 1 or <1. An RLP of 1 indicates that deviations of the respective laboratory are on average. Figure 3 shows z_U_ scores and the respective combination scores.

RSZ values of laboratories 4, 7, and 12 indicate a systematic positive bias (RSZ > +2) for the determination of the roe deer content, and RSZ values of the laboratories 1, 11, and 14 indicate a systematic negative bias (RSZ < −2). In fact, the vast majority of z_U_ scores were positive for the laboratories 4, 7, and 12, and negative for laboratories 1, 11, and 14. The z_U_ scores of laboratory 11 are particularly noticeable as results that were significantly too low (z_U_ score < −2) were obtained for five samples (3, 4, 5, 6, and 9). Interestingly, the lowest roe deer contents for all meat samples were determined by laboratory 11. With 111.56%, the amplification efficiency of the roe deer real-time PCR assay was considerably higher than 100%. However, with 91.98%, the amplification efficiency of the reference real-time PCR assay was much lower. These differences in the amplification efficiency explain why the roe deer content of the meat samples was systematically underestimated by laboratory 11.

## 4. Conclusions

Results obtained in the interlaboratory ring trial demonstrate the applicability of the real-time PCR assay for the detection and quantification of roe deer in meat samples to detect food adulteration. For none of the meat samples, false negative or false positive results were obtained. In ten out of eleven meat samples, the roe deer content was determined with satisfactory reproducibility and accuracy. Only for a heat-treated meat mixture, the roe deer content was ~five-fold overestimated. Overestimation of the roe deer content can be explained by differences in the amplifiability of the reference sequence compared to the roe deer specific sequence. A reference system published recently [43], amplifying a 70 bp fragment, is most probably more suitable for heat-treated products. This method has been successfully validated for the detection of animal components in vegan products by the Federal Office of Consumer Protection and Food Safety [45]. However, the applicability of the reference real-time PCR assay targeting a 70 bp fragment remains to be investigated in a further ring trial. Since heat-treatment procedures are known to affect DNA differently, the ring trial should include a variety of heat-treated model food products, e.g., brewed, cooked, and microwave treated ones.

## Figures and Tables

**Figure 1 foods-10-02645-f001:**
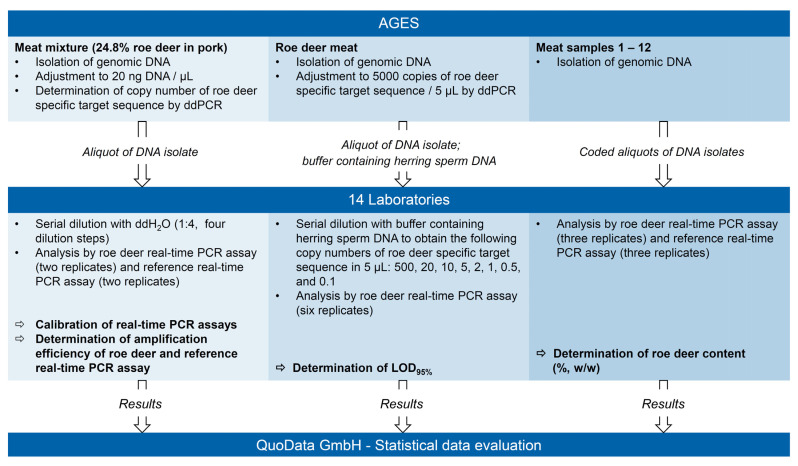
Design of the interlaboratory ring trial.

**Figure 2 foods-10-02645-f002:**
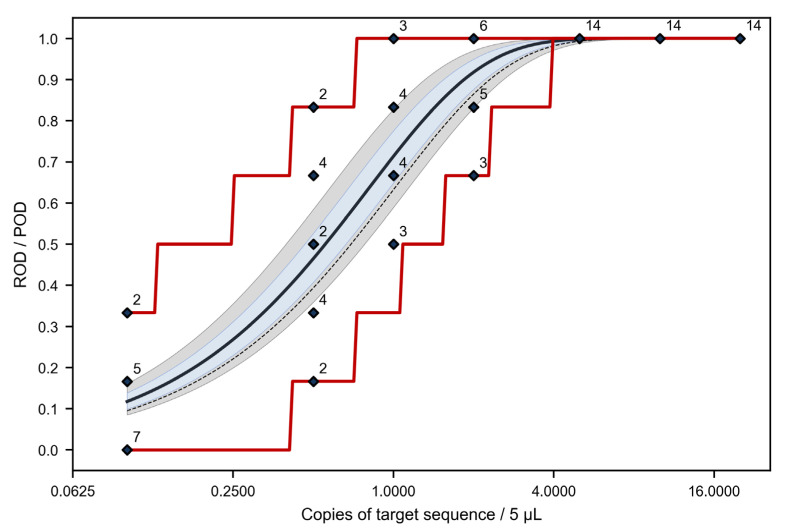
POD curve across laboratories (dark blue) with associated 95% confidence range (light blue) and 95% prediction range (light gray), laboratory-specific rate of detection (ROD) (blue diamonds, numerical values give the numbers of laboratories with the respective ROD) with associated 90% prediction interval (red). The ideal POD curve obtained under optimal conditions is given as dashed line.

**Figure 3 foods-10-02645-f003:**
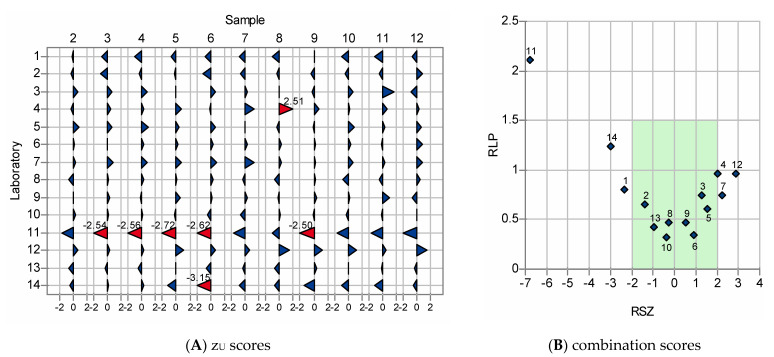
z_U_ scores (**A**) and combination scores (**B**) for the determination of the roe deer content in meat samples. In (**A**), z_U_ scores within −2 and +2 are shown in blue, z_U_ scores <−2 or >+2 in red. The figure includes laboratories which were identified as outliers.

**Table 1 foods-10-02645-t001:** Meat samples.

Meat Sample	Sample Name	Proportion of Roe Deer (%, *w*/*w*)
1	meat mixture 1	0
2	meat mixture 2	1
3	meat mixture 3	4.9
4	meat mixture 4	9.5
5	meat mixture 5	24.8
6	meat mixture 6	37.2
7	meat mixture 7	49.4
8	meat mixture 8	25.1
9	meat mixture 9, boiled	24.9
10	model sausage, raw	21.0
11	sausage, brewed	unknown ^1^
12	sausage, raw	unknown ^1^

^1^ declared to contain 5–10% (*w*/*w*) roe deer.

**Table 2 foods-10-02645-t002:** Primers and probes.

Assay	Primer/Probe	Sequence (5′-3′) ^1^	Final Concentration [nM]	Reference
	primer f	TGGCTGCTGCGTGCAGAA	200	
roe deer	primer r	TCTAAAATGCTTGGGAACCAGATAT	200	[14]
	probe	FAM-GAAGGGTCTCCGTCTGC-MGBNFQ	100	
	primer f	TTGTGCARATCCTGAGACTCAT	200	
myostatin	primer r	ATACCAGTGCCTGGGTTCAT	200	[22]
	probe	FAM-CCCATGAAAGACGGTACAAGRTATACTG-BHQ1	100	

^1^ FAM: 6-carboxyfluorescein, MGBNFQ: minor groove binding non-fluorescent quencher, BHQ1: black hole quencher 1, R: A + G.

**Table 3 foods-10-02645-t003:** Slope of the laboratory-specific standard curve, coefficient of determination (R^2^), and amplification efficiency (E) for the roe deer real-time PCR assay and the reference real-time PCR assay.

Laboratory	Roe Deer Real-Time PCR			Reference Real-Time PCR		
	Slope	R^2^	E (%)	Slope	R^2^	E (%)
1	−3.5759	0.9966	90.39	−3.3792	0.9992	97.66
2	−3.4180	0.9963	96.14	−3.3263	0.9968	99.82
3	−3.5573	0.9969	91.03	−3.5396	0.9985	91.65
4	−3.6037	0.9961	89.45	−3.6082	0.9970	89.30
5	−3.5877	0.9942	89.99	−3.3749	0.9999	97.83
6	−3.4349	0.9987	95.49	−3.3983	0.9937	96.91
7	−3.4698	0.9973	94.18	−3.4947	0.9989	93.26
8	−3.5276	0.9970	92.08	−3.3273	0.9978	99.78
9	−3.2937	0.9981	101.19	−3.3817	0.9996	97.56
10	−3.5797	0.9989	90.26	−3.4860	0.9996	93.58
11	−3.0728 ^1^	0.9986	111.56 ^2^	−3.5304	0.9615 ^2^	91.98
12	−3.4141	0.9980	96.29	−3.3037	0.9994	100.77
13	−3.4531	0.9975	94.80	−3.2621	0.9991	102.56
14	−3.4324	0.9986	95.59	−3.3850	0.9981	97.43

^1^ outlier (Grubbs test, α = 0.05), ^2^ outlier (Grubbs test, α = 0.01).

**Table 4 foods-10-02645-t004:** Laboratory-specific number of positive results obtained for the DNA isolate from roe deer meat, diluted to 20 to 0.1 copies of the roe deer-specific target sequence per 5 µL. A result was considered positive in cases in which the Ct value was <45 and the copy number, calculated based on the standard curve, was >0.

Laboratory	Copy Number/5 µL						
	0.1	0.5	1	2	5	10	20
1	0	2	5	4	6	6	6
2	2	4	5	5	6	6	6
3	0	5	4	6	6	6	6
4	0	3	4	4	6	6	6
5	2	4	3	6	6	6	6
6	0	1	3	4	6	6	6
7	1	3	6	6	6	6	6
8	1	2	5	5	6	6	6
9	0	5	6	5	6	6	6
10	0	4	5	5	6	6	6
11	1	1	4	6	6	6	6
12	1	2	6	5	6	6	6
13	0	4	3	6	6	6	6
14	1	2	4	6	6	6	6

**Table 5 foods-10-02645-t005:** Summary of results obtained for determination of LOD_95%_ of the roe deer real-time PCR assay.

Theoretical Copy Number of the Roe Deer-Specific Target Sequence Per 5 µL	Roe Deer Real-Time PCR			
	Number of Positive Tests/Total Number of Tests	Percentage of Positive Tests (%)	*p*_U_ (%) ^1^	*p*_O_ (%) ^2^
20	84/84	100.0	96.5	100.0
10	84/84	100.0	96.5	100.0
5	84/84	100.0	96.5	100.0
2	73/84	86.9	79.3	92.5
1	63/84	75.0	66.0	82.6
0.5	42/84	50.0	40.5	59.5
0.1	9/84	10.7	5.7	18.0

^1,2^ Clopper–Pearson confidence intervals. *p*_U_: lower limit of the 90% confidence interval for the detection probability *p*, *p*_O_: upper limit of the 90% confidence interval for the detection probability *p.*

**Table 6 foods-10-02645-t006:** Summary of the POD statistics for the real-time PCR assay for roe deer.

Parameter	Value
number of participating laboratories	14
number of PCR replicates per dilution level	6
model parameters of the POD curve:	
average amplification probability λ_o_	1.25
95% confidence interval for the estimated value of λ_o_	1.05–1.49
estimated value for slope b	1
laboratory standard deviation σ_L_	0.15
LOD_95%_ for median laboratory (copy number of the target sequence per 5 µL)	2.4

**Table 7 foods-10-02645-t007:** Summary of the statistical parameters for determining the roe deer content in meat samples. The distribution of readings for sample 8 deviates statistically significantly from the normal distribution. Therefore, the results for this sample have limited applicability.

Parameter ^1^	Sample 2	Sample 3	Sample 4	Sample 5	Sample 6	Sample 7	Sample 8	Sample 9	Sample 10	Sample 11	Sample 12
participating labs	14	14	14	14	14	14	14	14	14	14	14
labs with quantitative results	14	14	14	14	14	14	14	14	14	14	14
outlier labs	0	0	0	0	1	0	1	0	0	0	1
labs for determining parameters	14	14	14	14	13	14	13	14	14	14	13
theoretical value (%)	1.0	4.9	9.5	24.8	37.2	49.4	25.1	24.9	21.0	-	-
mean ± confidence level (%)	1.05 ± 0.07	5.22 ± 0.42	9.90 ± 0.76	24.02 ± 3.66	37.50 ± 3.19	47.40 ± 4.39	28.46 ± 1.93	132.57 ± 12.91	17.72 ± 1.57	7.63 ± 0.96	8.99 ± 0.86
s_R_ (%)	0.20	0.90	1.63	7.26	6.34	9.44	3.80	26.85	3.51	1.91	1.68
s_R rel_	19.50%	17.26%	16.46%	30.22%	16.91%	19.91%	13.35%	20.25%	19.83%	25.08%	18.71%
R (%)	0.57	2.52	4.56	20.33	17.76	26.42	10.64	75.18	9.84	5.36	4.71
R _rel_	54.59%	48.33%	46.09%	84.63%	47.36%	55.75%	37.38%	56.71%	55.52%	70.23%	52.40%
s_r_ (%)	0.19	0.56	0.98	2.93	3.29	5.71	1.88	14.36	2.35	0.83	0.82
s_r rel_	17.71%	10.63%	9.87%	12.20%	8.76%	12.05%	6.60%	10.83%	13.24%	10.83%	9.09%
r (%)	0.52	1.56	2.73	8.21	9.20	16.00	5.26	40.21	6.57	2.32	2.29
r _rel_	49.60%	29.77%	27.62%	34.17%	24.54%	33.75%	18.49%	30.33%	37.08%	30.34%	25.46%
recovery (%)	105.1	106.6	104.2	96.9	100.8	95.9	113.4	532.4	84.4	- ^2^	- ^2^

^1^ s_R_: reproducibility standard deviation; s_R rel_: relative reproducibility standard deviation; R: reproducibility limit, R _rel_: relative reproducibility limit; s_r_: repeatability standard deviation; s_r rel_: relative repeatability standard deviation; r (%): repeatability limit; r _rel_: relative repeatability limit. ^2^ Recovery could not be determined because the exact roe deer content was unknown.

## Data Availability

The datasets generated during the current study are available from the corresponding authors on reasonable request.

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
