# Peer review of "Real-Time PCR Assay for the Detection and Quantification of Roe Deer to Detect Food Adulteration—Interlaboratory Validation Involving Laboratories in Austria, Germany, and Switzerland"

_foods, 2021, doi:10.3390/foods10112645_

Round 1

Reviewer 1 Report

The Authors presented a work focused on interlaboratory validation, aiming at harmonizing analytical methods for food authentication across EU Member States. They tested a real-time PCR assay for roe deer in an interlaboratory ring trial including 14 laboratories from Austria, Germany, and Switzerland.

The Introduction section is precise and exhaustive and it well describes the state of the art and the scope of the manuscript.

Materials and methods section is correctly reported.

I would add information about 1) sample storage before running the analyses and 2) time before running the analyses.

DNA degradation can be a relevant issue, especially when quantification from complex matrices occurs. Thus, these details should be added to the methods section. 

Results and Discussion section is clearly stated and figures are clear. 

L297-301 should be moved to Methods.

Conclusion section is concise and it is well supported by the results obtained.

Reviewer 2 Report

Comments to the Author
This manuscript is the natural conclusion of the research group's work on a collaborative study to validate a method for quantifying game meat in food products.

The work is presented in a very clear and comprehensive manner. The way in which the study was organised and the analysis of the data is largely in line with international standards. However, there are a couple of issues that should be addressed.Critical points
1.      in the materials and methods section, in particular section 2.2. Design of the interlaboratory ring trial, some passages use the expression "should be".

The impression is that those parts were directly taken from the protocol provided to the participants of the collaborative trial without further elaboration of the text. The authors should make these parts more homogeneous with the rest of the section by using the past tense.
2.      Again in the materials and methods section, section 2.5. Real-time PCR, the description of the PCR protocol states the following: "Real-time PCR reactions were performed in a total volume of 25 µL, consisting of

20 µL of reaction mix and 5 µL of isolated DNA. The following temperature program was applied for both the roe and reference real-time PCR assays: 15 min at 95 °C, 40 cycles of 60 s at 94 °C and 60 s at 60 °C'. If indeed the amplification protocol involves 40 cycles it is not clear how it is possible that, as stated in section 3.2. Level of detection (LOD95%), "A result was considered positive in case the Ct value was < 45". The authors should resolve this discrepancy.

Reviewer 3 Report

The study investigated the interlaboratory validation of real-time PCR assay for the detection and quantification of roe deer to detect food adulteration. The study appears promising, however and in reviewer's opinion, there are quite a number of concerns that need to be addressed:
- The title needs to be modified, reviewer suggests : "Real-time PCR assay to detect and quantify food adulteration of roe deer meat samples: An interlaboratory validation involving Austria, Germany, and Switzerland" ...please kindly modify the abstract, to reflect this.

-Introduction:
Please, introduction, paragraph 4, needs more information about the importance of validation process in food safety (https://doi.org/10.1080/87559129.2021.1938600), why food adulteration must be tackled (for example, Journal of Food Bioactives 2019, 6: 6-9), before mentioning real-time PCR assay as a detection method. The references provided are very useful for this work, and highly recommended.

-Materials and methods
This is a bit ok but needs more work. It will be very helpful for readers, if authors can prepare a picture of europe, and spot the location of these laboratories. This should be the figure 1, I believe authors can do this very easily. Secondly, for each location, insert in bracket the GRPS/GPS or latitude longitude location coordinates. This will help to make it more authentic, and very valid.
Table 1, please clearly explain how meat sample has been derived. This is not clear to the reader. How did you prepare these samples? Yes, figure 1 shoes a bit of it, but it is not clear. describe it step by step into its detail Please provide more information about the process of conducting the Real-time PCR. Line 152-164 is not enough, more details please to guide learners and enhance reproducibility .

- Results and discussion
This section is very good. table 3, laboratory 1-14 refers to each laboratory mentioned in materials and methods. Therefore it has to be shown in the key of Table 3, so show lab 1=...; lab 2=...., ...lab 14 =...
Please, do the same for Table 4 key, so that readers are properly guided.
Please, the results and discussion section is very scanty, there is a lot of data description, but not data discussion. Please, authors, do a literature search, and extract more relevant literature that brings out a better understanding of your work.
Also, in your results, provide trends of how the various sample locations follow in the detection outcomes, apply your discretion to choose the parameter of interest, and be able to justify that parameter trend (by location, in descending or ascending order) with relevant literature
- Conclusion seems ok, however, the authors need to do more reflection: was the objective of this study fully actualized? are there any limitations? what are your recommendations for future studies? Please kindly brainstorm and offer more pragmatic insights ok

Round 2

Reviewer 3 Report

Thank you authors for your responses . 
The reviewer opines that, while authors may have a point in the manner in which they have responded to the queries reviewer has raised, it should be born in mind that authors are not presenting their work only for themselves, but to the greater public. And therefore having only one mindset defeats the purpose of broadening the readership of their own work.

The reviewer therefore encourages the authors to rethink and revisit the questions raised, and do justice to them, rather than clearly rejecting them, less an unnecessary struggle between who is right or who is wrong ensues.

The title needs to be modified as suggested, among other things.

Author Response

Response to Reviewer 3

  • As suggested by the reviewer, we changed the title.

  • We added more information on the importance of method validation, as proposed by the reviewer.

  • The reviewer suggested including a picture of Europe, showing the location of the laboratories that participated. Actually, we do not understand the benefit of including a map of Europe, showing the location of the laboratories. The same holds true for GRPS/GPS or latitude longitude location coordinates. In our opinion, this would not “help to make” the study “more authentic, and very valid”.

  • As suggested by the reviewer, we explained preparation of the samples in more detail.
  • The reviewer suggested “providing more information about the process of conducting the Real-time PCR” “to guide learners and enhance reproducibility”. We checked again the paragraph describing experimental parameters for performing real-time PCR. Actually, the paragraph contains all the information required for applying both the specific real-time PCR assay for roe deer and the reference PCR assay, including sequences and concentrations of primers and probes, PCR kit, and temperature protocol.
  • According to the reviewer, table 3 should not only contain numbers of the laboratories but also indicate the names of the institutions. The aim of an interlaboratory ring trial is to evaluate the overall performance of a specific method in different laboratories, but not to assess the performance of individual laboratories (like in proficiency testing). Thus, in our opinion, it is not necessary to disclose which laboratory has number 1, 2 etc.
  • As suggested by the reviewer, we discussed the results in more detail.
  • The reviewer suggested to “provide trends of how the various sample locations follow in the detection outcomes, apply your discretion to choose the parameter of interest, and be able to justify that parameter trend (by location, in descending or ascending order) with relevant literature.” Actually, the meaning of this paragraph is not clear at all. What does the reviewer mean with “various sample locations”? As outlined in the methods section, all samples were prepared at the AGES. What does the reviewer mean with “Apply your discretion to choose the parameter of interest” and “be able to justify that parameter trend (by location, in descending or ascending order) with relevant literature”?
  • As suggested by the reviewer, we extended the Conclusion section.